# Development of a novel startle response task in Duchenne muscular dystrophy

Kate Maresh[1,2], Andriani Papageorgiou[1], Deborah Ridout[3,4], Neil Harrison[5], William Mandy[6], David Skuse[7], Francesco Muntoni[1,2,4]*

1 Dubowitz Neuromuscular Centre, UCL Great Ormond Street Institute of Child Health, London, United Kingdom, 2 MRC Centre for Neuromuscular Diseases, UCL, London, United Kingdom, 3 Department of Population, Policy & Practice, UCL Great Ormond Street Institute of Child Health, London, United Kingdom, 4 NIHR Great Ormond Street Hospital Biomedical Research Centre, Great Ormond Street Institute of Child Health, University College London, & Great Ormond Street Hospital Trust, London, United Kingdom, 5 Division of Psychological Medicine and Clinical Neurosciences, Cardiff University, Cardiff, United Kingdom, 6 Department of Clinical, Educational and Health Psychology, UCL, London, United Kingdom, 7 Department of Behavioural and Brain Sciences, UCL Great Ormond Street Institute of Child Health, London, United Kingdom

* f.muntoni@ucl.ac.uk

**Data Availability Statement:** All relevant data are available within the paper or on request to preserve the anonymity of participants given the rarity of DMD. Approving Ethics Committee: NHS Health

## Abstract

Duchenne muscular dystrophy (DMD), an X-linked childhood-onset muscular dystrophy caused by loss of the protein dystrophin, can be associated with neurodevelopmental, emotional and behavioural problems. A DMD mouse model also displays a neuropsychiatric phenotype, including increased startle responses to threat which normalise when dystrophin is restored in the brain. We hypothesised that startle responses may also be increased in humans with DMD, which would have potential translational therapeutic implications. To investigate this, we first designed a novel discrimination fear-conditioning task and tested it in six healthy volunteers, followed by male DMD (n = 11) and Control (n = 9) participants aged 7–12 years. The aims of this methodological task development study were to: i) confirm the task efficacy; ii) optimise data processing procedures; iii) determine the most appropriate outcome measures. In the task, two neutral visual stimuli were presented: one 'safe' cue presented alone; one 'threat' cue paired with a threat stimulus (aversive noise) to enable conditioning of physiological startle responses (skin conductance response, SCR, and heart rate). Outcomes were the unconditioned physiological startle responses to the initial threat, and retention of conditioned responses in the absence of the threat stimulus. We present the protocol development and optimisation of data processing methods based on empirical data. We found that the task was effective in producing significantly higher physiological startle SCR in reinforced 'threat' trials compared to 'safe' trials (P < .001). Different data extraction methods were compared and optimised, and the optimal sampling window was derived empirically. SCR amplitude was the most effective physiological outcome measure when compared to SCR area and change in heart rate, with the best profile on data processing, the least variance, successful conditioned response retention (P = .01) and reliability assessment in test-retest analysis (rho = .86). The definition of this novel outcome will allow us to study this response in a DMD population.

Research Authority, London Bridge Research Ethics Committee; Reference no. 18/LO/1575. Non-author institutional point of contact for field data access queries: Dr. Marta Zancolli, Senior Research Coordinator (m.zancolli@ucl.ac.uk). Developmental Neurosciences Research & Teaching Department, UCL Great Ormond Street Institute of Child Health, 30 Guilford Street, London WC1N 1EH.

**Funding:** The study 'A Study of Emotional Function in Duchenne Muscular Dystrophy (EmoDe Study)' was funded by Great Ormond Street Hospital Children's Charity (Award no. V0119), and supported KM and AP (https://www.gosh.org/). KM was also supported by the Medical Research Council (MRC) via the MRC Centre for Neuromuscular Diseases Grant no. MR/K000608/1 (Institute of Neurology, University College London; https://www.ucl.ac.uk/centre-for-neuromuscular-diseases/). The partial support of EU Horizon 2020 Grant no. 83245 Brain Involvement iN Dystrophinopathies to FM (UCL) is also acknowledged (https://ec.europa.eu/programmes/horizon2020/en/h2020-sections-projects). The funders had no role in study design, data collection and analysis, decision to publish, or preparation of the manuscript.

**Competing interests:** I have read the journal's policy and the authors of this manuscript have the following competing interests: FM has received grants, speaker and consultancy honoraria from Sarepta Therapeutics, Avexis, PTC Therapeutics, Roche, Biogen, Dyne Therapeutics, Novartis and Pfizer; DS has current grant funding from the Medical Research Council (UK), Sarepta Therapeutics, the European Research Council, and the Patrick Paul Foundation; WM is supported by the National Institute of Health Research, Autistica, the Dunhill Medical Trust, the Medical Research Council and the European Research Council; NH has received consulting fees from Janssen and GSK. The remaining authors have declared no competing interests.

**Abbreviations:** ASD, Autism spectrum disorder; CS, Conditioned stimulus; DMD, Duchenne muscular dystrophy; ECG, Electrocardiogram; EDA, Electrodermal activity; EEG, Electroencephalography; fMRI, Functional magnetic resonance imaging; GABA, Gamma amino butyric acid; GOSH, Great Ormond Street Hospital for Children; HR, Heart rate; ITI, Inter-trial interval; LOA, Limits of agreement; MEG, Magnetoencephalography; MRI, Magnetic resonance imaging; SCL, Skin conductance level; SCR, Skin conductance response; TTP, Time-to-peak; UCL, University College London; UCL GOS

## Introduction

Duchenne muscular dystrophy (DMD) is a childhood-onset, progressive, life-limiting muscle-wasting disorder, which is frequently associated with complex neuropsychiatric co-morbidities [1]. It is an X-linked disorder caused by mutations in the *DMD* gene, which lead to a loss of the protein dystrophin [2]. Boys with DMD have long been noted to have a higher prevalence of intellectual disability than the typical population [3, 4]. There is also a higher prevalence of other neurodevelopmental disorders such as autism spectrum disorder, attention deficit hyperactivity disorder and emotional and behavioural problems than the typical population [5–9]. Whilst several studies have found clinically significant anxiety or internalising symptoms in 23–27% of the DMD participants studied [6–8, 10], anxiety symptoms in DMD have received relatively little attention.

The neurobiological basis of anxiety disorders has been extensively investigated in typically developing populations. The 'emotion' circuits of the brain involve the amygdala and other deep brain nuclei, as well as connected cortical structures often known together as the limbic system [11]. Dystrophin is highly expressed in the amygdala, hippocampus and other limbic structures in humans [12]. Several studies have implicated changes in limbic structures resulting from a lack of dystrophin in the pathogenesis of the neuropsychiatric phenotype in DMD. Studies in a mouse model of DMD, the *mdx* mouse, have shown reduced Gamma amino butyric acid-A ($GABA_A$) receptors in the hippocampus and amygdala [13, 14]. In DMD patients there is reduced $GABA_A$ receptor distribution in the prefrontal cortex, which closely interconnects with limbic structures [15]. A recent study of resting-state functional magnetic resonance imaging (MRI) found hyperconnectivity in the default-mode network connections between some limbic structures, including the hippocampus, posterior cingulate and precuneus in the DMD group compared to controls [16]. Furthermore, *mdx* mice show a behavioural phenotype implicating emotion circuits, demonstrating exaggerated unconditioned 'freezing' startle responses compared to wild-type mice when exposed to a threat stimulus, which can be normalised with treatment to restore functional dystrophin [13, 17]. Therefore, it may be that pathological changes in the emotion circuits due to a lack of dystrophin leads to an increase in anxiety symptoms in DMD, although this has not previously been investigated in humans with DMD.

Anxiety in humans is thought to be caused by reactivation of innate fear circuits, and that excessive activation in these circuits could cause pathological anxiety [18]. The neurobiology of fear and startle responses has been extensively investigated with behavioural studies in animals and humans using cued fear-conditioning experimental paradigms [19–21]. These paradigms use classical fear conditioning to produce associative fear-learning, by pairing a neutral stimulus (conditioned stimulus, CS) with an aversive 'threat' stimulus, known as the unconditioned stimulus (UCS), for example a loud noise or low amplitude electric shock. The UCS elicits a behavioural and/or physiological startle or fear response (Fig 1A). Repeated presentation of this pairing leads to a learned association or 'conditioning' of the physiological responses to the neutral CS, so subsequent presentation of the CS elicits this conditioned response even without the 'threat' UCS. This learned response can subsequently be extinguished with repeated presentation of the CS without the UCS. In discrimination fear conditioning paradigms, two neutral CS are used: a 'threat' cue, which is conditioned by the UCS as above (CS+), and a 'safe' cue (CS-), which enables comparison of the responses to both cues [22].

Individuals with anxiety disorders have been extensively studied with classical fear conditioning paradigms [23]. Physiological startle responses to aversive stimuli in fear-conditioning tasks are typically greater in anxious compared to non-anxious individuals. The degree of

ICH, UCL Great Ormond Street Institute of Child Health.

acquisition and extinction of conditioned startle responses amongst anxious and non-anxious subjects has varied across paediatric studies, with some showing larger and more persistent conditioned responses in extinction in anxious vs. non-anxious children, and others showing no difference [23–25].

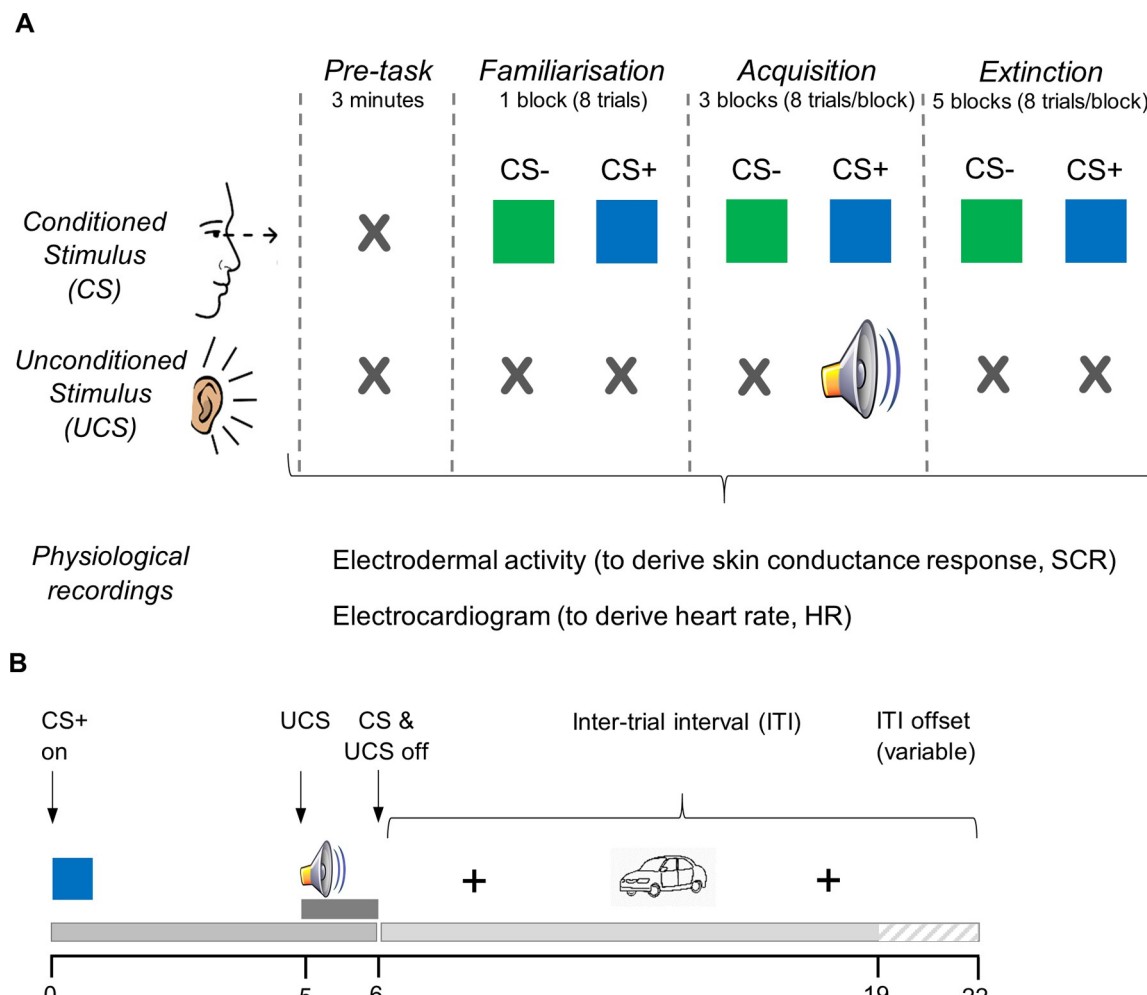

**Fig 1. Schematic diagrams of the discrimination fear conditioning task and individual trial protocols.** (**A**) Trials of two different neutral visual stimuli (coloured squares) called the conditioned stimuli (CS) are presented to the subject in pseudo-randomised trials. One is a 'threat' cue (CS+), which is paired with the unconditioned stimulus (UCS), an aversive noise, in the Acquisition phase only. A 'safe' cue (CS-) is presented alone with no UCS. The 'threat' CS+ and 'safe' CS- are presented in equal proportions throughout all phases in a pseudo-randomised order. Each block comprises 8 trials, with on average four of each CS per block. In the pre-task calibration phase, no stimuli are presented to allow for baseline skin conductance (SCR) and heart rate (HR) recordings and calibration physiological manoeuvres. The Familiarisation phase (one block) comprises eight CS+/CS- trials presented with no UCS. In the Acquisition phase (three blocks), 10/12 CS+ trials are paired with the aversive UCS (loud white noise) presented binaurally via headphones (~80% reinforcement schedule). After 1–2 hours the Extinction phase is conducted (five blocks), comprising 40 CS+/CS- trials with no paired aversive noise. SCR and HR responses are recorded throughout. (**B**) Structure of a CS+ trial showing the onset and offset times of conditioned (CS) and unconditioned (UCS) stimuli, and the inter-trial interval (ITI). Each CS+/CS- stimulus is presented for 6s, with onset of UCS after 5s (duration 1s). Both CS and UCS co-terminate 6s into the trial. The ITI begins with a fixation cross for 4s, followed by a neutral picture (black and white drawings of common objects: e.g. house, shoe and car) to maintain attention during the ITI, and ends with a fixation cross prior to commencement of the next trial. The ITI duration is randomised between 13-16s. CS- trials and unreinforced CS+ trials are identical, except for the omission of the UCS. Break periods of 20s occurred between each block of 8 trials.

Given the increased prevalence of anxiety in DMD and the potential for underlying pathological differences in the emotion circuits in DMD, we aimed to design a novel fear-conditioning task that could be used to investigate startle responses in young males with DMD. These tasks have been used before in paediatric populations, but not in young people with DMD. Given the complex neuropsychiatric profile affecting some people with DMD, there are a number of methodological aspects that are important to take in consideration when developing such a task for this specific population. Indeed, some boys with DMD have sensory processing difficulties, including sensory over-responsivity to loud sounds [26]. Sensory over-responsivity can be associated with autism spectrum disorder (ASD) and other neurodevelopmental disorders, which are common DMD comorbidities, and there is some evidence that it is linked to changes in inhibitory GABAergic synapses [27, 28]. Most previous studies in children have used auditory stimuli as the UCS, at sound levels ranging from 83–106 dB, rather than an electric shock stimulus (see S1 File) [29–31], so the sound level of auditory stimulus would need to be appropriate for participants with possible sensory over-responsivity.

Tolerability and drop-out has been a problem in some paediatric studies with more aversive unconditioned stimuli, such as a fearful face paired with a screaming sound, which had drop-out rates of between 10–42% in the non-anxious groups [32–34]. Other studies have shown that conditioning can be enhanced by using more salient conditioned stimuli, such as a fearful face as the conditioned stimulus rather than a neutral shape [35] however, this could be a confounding factor in the context of DMD, as impaired facial emotion recognition can be associated with the condition [36].

Fear-conditioning studies have employed a number of different physiological outcome measures, which arise from autonomic nervous system reactivity to a 'threat' presentation and subsequent conditioned responses. The skin conductance response (SCR), a measure of skin sweating, is a well-established and reliable marker of sympathetic activity in fear-conditioning paradigms and is commonly used in both paediatric and adult studies [24, 29, 37–40]. Other measures used in paediatric studies include fear-potentiated startle and self-reported fear; heart rate and heart rate variability have also been used in adult fear-conditioning studies [39, 41, 42].

Heart rate (HR) can either decrease or increase depending on the balance between parasympathetic and sympathetic activity: a short-lasting parasympathetically-mediated bradycardia occurs in both humans and animals in response to threat, typically associated with the freezing response in animals, whilst sympathetically mediated tachycardia arises with conditioning to a given context [42–44]. In DMD there is typically a raised resting heart rate and decreased heart rate variability [45, 46], however using HR relative to the baseline may be a useful outcome.

The eyeblink reflex in the measurement of fear-potentiated startle measured with electromyography of the orbicularis oculi muscle is another measure of the initial startle response [47], although this measure could be confounded in the DMD population due to the underlying muscle pathology. Pupillary responses, electroencephalography (EEG), magnetoencephalography (MEG) and functional magnetic resonance imaging (fMRI) have also been used in studies in the typical population [48–50].

In this protocol-development study we aimed to i) confirm the efficacy of a newly developed fear-conditioning task; ii) optimise data processing procedures; iii) determine the most appropriate outcome measures for the startle response. We present novel fear-conditioning task protocol and its empirical evaluation, firstly in a small group of healthy volunteers, and then in a group of 20 young male DMD and age-matched Control participants. The definition of this novel outcome measure will then allow us to study a DMD patient population for this response.

## Materials and methods

### Recruitment

For the first stage of this task development study, we recruited six healthy volunteers, via advertising at University College London (UCL) UCL Great Ormond Street Institute of Child Health (UCL GOS ICH) from September 2018 to January 2019. Inclusion criteria were: aged 8–25 years; no known neuromuscular disease, mental health diagnosis, cognitive impairment or uncorrected auditory or visual impairment. The study was approved by the UCL Research Ethics Committee (REC) (Project ID no. 9639/001) and was conducted at UCL Great Ormond Street Institute of Child Health. All participants were provided with participant information leaflets in advance and gave written, informed consent or assent prior to the study.

For the second task development stage, we recruited 20 male participants aged 7–12 years, half DMD and half age-matched control participants from February to May 2019, with follow-up visits for DMD participants after three months (May to August 2019). Inclusion criteria were: male; age 7–12 years; no significant uncorrected auditory/visual impairment or noise-sensitivity; no neurological/psychiatric diagnosis (Controls only); DMD only—genetically-proven diagnosis of DMD and not receiving ataluren (a dystrophin-modulating therapy targeting nonsense mutations that has the potential to cross the blood-brain barrier). Participants were also asked about visual impairments, including colour-blindness, and whether they experience significant sensory over-responsivity to loud sounds, in which case taking part in the study was advised against.

The narrower age range, limiting to pre-pubertal participants, was used for the second stage of the study with DMD and Control participants, as there is evidence in both mouse and humans that age of subject can affect fear responses, with increased responses in adolescence compared to childhood and adulthood [22, 29, 33]. Pubertal status was assessed by parents and/or participants using the 'Growing and Changing' questionnaire [51].

Ethical approval was granted by the Health Research Authority (HRA), London Bridge Research Ethics Committee (18.LO.1575). The study assessments took place at Great Ormond Street Hospital for Children (GOSH) and UCL GOS ICH. All participants and their parents/guardians were provided with age-appropriate participant information leaflets prior to the study and gave their written informed assent or consent respectively.

### Fear-conditioning task protocol

**Task design.** We designed a discrimination fear-conditioning paradigm, which involves two conditioned stimuli (CS) (Fig 1A). One was the 'threat' stimulus (CS+) which cued the aversive unconditioned stimulus (UCS), and the other the 'safe' stimulus (CS-) which was not associated with the UCS. The CS+/CS- were coloured squares (green and blue) of similar tone and colour depth to make them as similar as possible and with limited prior connotations (e.g. not a typical 'danger/warning' colour such as red or orange). Faces were not used to avoid potential bias due to poorer facial effect recognition in some boys with DMD [36]. The CS + colour (either blue or green) was counterbalanced randomly between participants to avoid any stimulus bias.

The auditory aversive unconditioned stimulus (UCS) was a computer-generated white noise of duration 1s at a sound level of 85–90 dB, which is towards the lower end of that used in previous studies (S1 File) and was chosen as a balance aiming for a sufficiently aversive UCS to elicit physiological startle responses but not so aversive as to cause distress. As a comparison, 90dB is the sound level heard when using a hairdryer, lawnmower or blender [52].

The task was divided into four stages, detailed in Fig 1A. During the pre-task calibration period, physiological manoeuvres (deep breath; sudden head turn) and an external stimulus (a loud hand clap) were performed as calibration checks, and neutral images shown in the inter-trial intervals and break periods were presented so participants could habituate to these. In the Familiarisation phase, CS+/CS- were presented with no UCS, allowing participants to also habituate to the conditioned stimuli, and provide baseline measurements of responses to the CS+/CS- alone. In the Acquisition phase, CS+ was paired with the aversive UCS in a partial reinforcement schedule, with 10/12 CS+ trials being reinforced with the UCS and 2/12 unreinforced. Partial reinforcement schedules typically strengthen conditioning responses to the UCS [53]. The Extinction phase occurred after a break of at least 1–2 hours and comprised presentation of 40 CS+/CS- trials without the UCS. This enabled measurement of the degree of retention of the conditioned fear response to CS+, and the time taken to extinguish the conditioned response. Due to time constraints it was only feasible to perform the Extinction phase on the same day, although ideally a gap of 24 hours between acquisition and extinction is preferable for stronger retention of conditioned responses [54].

A schematic diagram of the structure of a trial is shown in Fig 1B, comprising presentation of one CS (CS+ or CS-), with or without the UCS, followed by a variable inter-trial interval (ITI) that allowed enough time for a return of physiological responses to baseline [38]. Simple neutral images were displayed on the screen for part of the ITI to maintain participants' attention, followed by a fixation cross prior to onset of the next trial. Break periods of 20s duration occurred between each block, during which simple written instructions and encouraging comments were displayed on the screen. Where participants found reading difficult, the researcher read out the displayed text.

Following the testing of the first two healthy volunteer participants, several changes were made to the task protocol. The trial timings had initially been longer (CS presentation 8 s, UCS 2 s, break periods 30 s) and were subsequently shortened as above to improve retention of attention, based on feedback from participants. To further improve attention, neutral images were presented during the ITI and the break periods. In the Extinction phase, the task initially comprised 24 trials (three blocks) based on similar previous studies, however preliminary data suggested that extinction of the conditioned response was not complete by the third block, therefore the Extinction phase was extended to five blocks (40 trials).

Test-retest reliability of similar paradigms has been established in healthy individuals [37], however as this was the first such study in a DMD population we invited DMD participants to return for a second visit after three months to repeat the fear-conditioning task to enable test-retest reliability assessment.

**Stimulus presentation and event recording.** The task was scripted in MATLAB (2017b, Mathworks) using the Psychophysics Toolbox extensions [55, 56]. The CS (blue and green squares), UCS (white noise) and text instructions were generated from code within the script. Additional images were presented within the script as.jpg or.gif files. Text instructions were kept as short as possible and in large font, as boys with Duchenne muscular dystrophy can have reading difficulties [57]. Digital event markers in the script indicated the start and end of trials, CS and UCS onset, and manually triggered markers for events that may cause non-stimulus-related physiological responses (e.g. cough, talking, deep breath, movement or an external noise). Digital events were recorded using a USB TTL event marking module (BlackBox Toolkit Ltd, Sheffield, UK) to accurately record the timing of these events within the physiological data recordings.

Participants undertook the task while seated in a quiet room, either in an outpatient clinic or a research facility. They were seated in front of a laptop computer screen wearing over-ear headphones. Participants were verbally instructed to maintain their focus on the screen and

keep still, and that the task included hearing loud sounds which may startle them, and that they could stop the test at any time. The researcher and parent/guardian were present but out of sight of the participant during testing.

**Physiological variable data acquisition and processing.**   The physiological variables recorded were the electrodermal activity (EDA), from which the skin conductance response (SCR) was derived, and the electrocardiogram (ECG). from which heart rate (HR) was derived.

Surface skin electrodes were applied to the subject 5–10 minutes before the start of testing to allow the electrode paste to be absorbed into the skin. Two disposable Ag/AgCl surface electrodes pre-filled with isotonic gel were placed on the palmar surface of the 2nd and 3rd digits of the non-dominant hand to record the EDA. Three standard limb ECG electrodes were placed on the torso and wrist. ECG and EDA leads were connected to a Biopac MP160 unit (Biopac Systems, Inc., Aero Camino, CA) and calibrated. EDA and ECG were recorded throughout all phases of the task. EDA data was recorded using a sampling rate of 2000 Hz and a SCR response threshold of 0.01μS [37, 38, 58], and ECG data was recorded at a sampling rate of 1000 Hz [59].

The sound level of the UCS was checked with an external digital sound level meter prior to each task. The ambient temperature of the room was recorded using a digital thermometer, as this can correlate positively with SCR level, when tested over a range from 20–40˚C affect SCR [60]. Most Control participants were tested in a research facility room (median temperature 21.0˚C; range 18.9–23.3˚C), whereas DMD participants were tested in several different clinical settings (median temperature 19.3˚C; range 18.3–22.5˚C). Whilst there was a significant difference between these values on Mann Whitney U-test comparisons ($U = 10.5$, $P = .04$), temperature did not correlate with either baseline SCR ($rho = 0.15$, $P = 0.58$) or $SCR_{uc}$ ($rho = -0.13$, $P = 0.66$). Linear regression analysis confirmed that temperature was not a significant predictor of either of these variables.

EDA recordings were re-sampled at 15.625Hz and filtered using a 1Hz low pass filter, following Biopac EDA guidelines, to rectify and smooth the data. ECG recordings were filtered using a band pass filter at 1–35 Hz, as per Biopac guidelines, and 2% noise rejection to optimise ECG complex peak detection [59]. ECG data was transformed into R-R intervals in seconds, the time intervals between each heartbeat (the 'R' indicating the R-wave, which is the maximal positive peak of the ECG complex).

**Empirical optimisation of sampling window.**   EDA data were initially extracted using an automated event-related analysis using a priori sampling windows of 0-10s after CS onset, determined by the timings of stimulus presentation and previously published data on the timings of the latency, peak and recovery time of skin conductance and heart rate responses [38, 40]. This was refined empirically by determining the 'time-to-peak' (TTP) using the latency and rise-times (time from SCR onset to peak), where TTP = latency+rise-time (Fig 2A), to ensure that the time to the SCR peak following UCS-onset time was adequately captured. The mean TTP was longer in the first CS+ trial compared to the remaining trials: mean TTP in the first CS+ trial 13.0s (sd 1.6), mean TTP in remaining trials 8.9s (sd 1.8). By taking the mean plus two standard deviations as an empirical sampling window to ensure maximal capture of outliers, the window would be 12.5s from CS-onset. However, a window >12s would increase the risk of NS-SCRs related to the start of the ITI being captured based on the trial timings. Therefore, to maximise peak detection and minimise the chance of non-specific SCRs being included, we redefined the data extraction window as 0-12s after CS-onset for our paradigm. Manual inspection was still performed for all trials, especially for the first SCR peak, to ensure any outliers were detected but the new window duration optimised peak detection. All EDA and ECG data were re-extracted using this new 0-12s window.

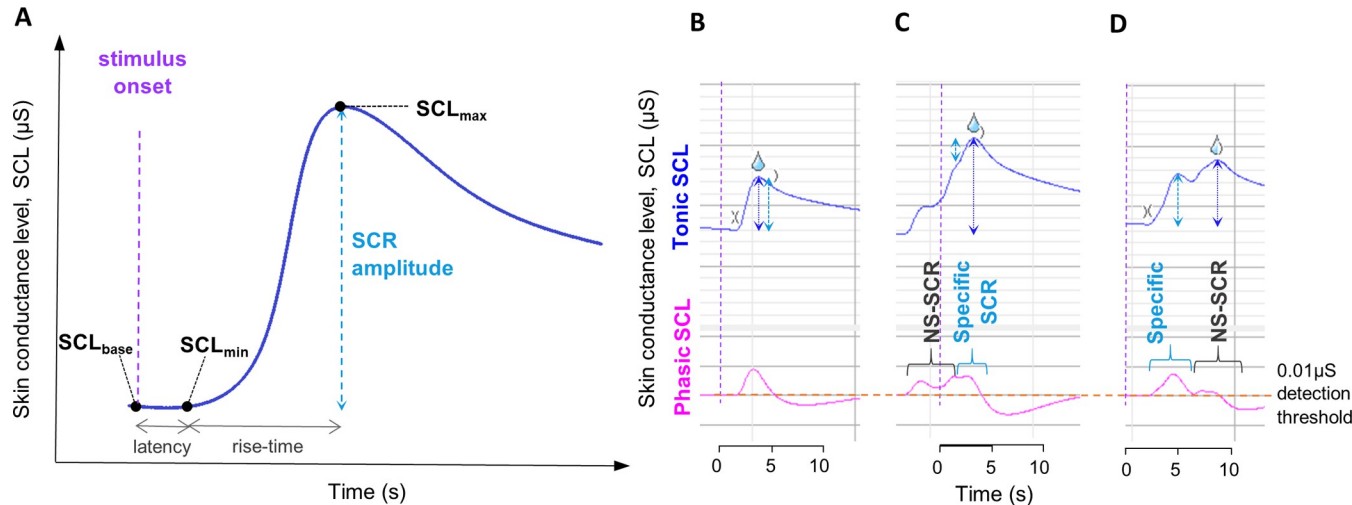

**Fig 2. Measurement of skin conductance responses.** (**A**) Schematic diagram of a typical skin conductance response following presentation of the conditioned stimulus, CS. The amplitude of the skin conductance response ($SCR_{amp}$) is measured from the baseline skin conductance level ($SCL_{base}$) to the maximum SCL at the peak of the response ($SCL_{max}$). If artefacts in the data affect $SCL_{base}$, the minimum SCL before the peak ($SCL_{min}$) can be used as a baseline measurement instead. Latency is the time from CS onset to the start of the SCR (defined as when the SCR crosses the 0.01μS detection threshold) and is typically 1-3s. Rise-time is the time from SCR onset to the peak response ($SCL_{max}$). The 'time-to-peak' (TTP) equals latency + rise-time. For the TTP calculations to determine the sampling window, latency was taken from the CS presentation time rather than the UCS (unconditioned stimulus) presentation time. (**B**)-(**D**): SCL data from a single participant analysed with automated event-related data extraction. Upper trace (blue) is the tonic skin conductance level (SCL) from which SCRs are measured. Lower trace (pink) is the phasic SCL, used to determine when threshold of 0.01μS is passed for event-related response identification in the automated analysis. Stimulus onset time (time = 0s) indicated by purple arrow/dashed purple vertical line. Orange dashed horizontal line on phasic SCL indicates the 0.01μS detection threshold. Open parenthesis symbol indicates the start of the SCR when the threshold is crossed. Closed parenthesis symbol indicates the end of the SCR when the phasic SCL drops below 0.01μS. 'Water drop' symbol denotes the event-related SCR determined by the automated analysis. Dark blue dotted vertical arrows indicate the automated amplitude measurement; Light blue dotted arrows indicate the actual SCR amplitude measurement. (**B**) A typical SCR, which has been correctly identified and measured. (**C**) A non-specific skin conductance response (NS-SCR) starts before CS is presented; a slight hump in the upward NS-SCR indicates where the event-related SCR starts but the automated response measurement (dark blue arrow) overestimates the SCR by measuring the combined amplitude of both responses. This is difficult to accurately re-measure manually so would need to be excluded. (**D**) A typical response starts but as the phasic response (pink line) does not re-cross the 0.01 μS threshold (orange dashed line) when a second response occurs (e.g. due to a physiological response from a deep breath) the peak of the second response is taken as SCLmax so the automated measurement overestimates the SCR. Responses such as these can be accurately re-measured manually.

**Data extraction and cleaning.** EDA data was extracted from Biopac AcqKnowledge 5.0.3 into Microsoft Excel, using two methods which were then compared: 1) an automated Event-related analysis to linking SCRs to digital event markers for CS+/CS-/UCS presentations; 2) an 'Epoch' analysis where pre-specified measurement parameters could be set for extraction within a given time window after the digital markers. The EDA is the absolute skin conductance level (SCL) in μS, from which the SCR amplitude ($SCR_{amp}$, in μS) was derived, the difference between skin conductance level ($SCL_{base}$) at start of the trial and at maximal response $SCL_{max}$, i.e. $SCR_{amp} = (SCL_{max}) — (SCL_{base})$, as shown in Fig 2A. The Epoch analysis parameters were: $SCL_{base}$, $SCL_{max}$, the minimum SCL in the extraction window ($SCL_{min}$), the area under the curve of the SCR ($SCR_{area}$, in μS-s). ECG data was also extracted using the Epoch analysis method: the mean R-R interval within the sampling window. Heart rate (HR) in beats per minute (bpm) was derived from the R-R interval using the calculation: HR = 60/R-R.

Data cleaning involved manual inspection of all SCRs, cross-referencing digital markers. Adjustments to automated measurements were made where the true specific SCR had been incorrectly measured due to artefacts or non-specific SCRs (NS-SCRs) or incorrectly assigned by the automated analysis. Non-specific SCRs (NS-SCR) are physiological responses related to an external or other internal physiological stimulus, or background skin conductance activity. Examples of the manual inspection process are shown in Fig 2. For the event-related analysis,

if a non-specific SCR (NS-SCR) started before CS presentation or just after the specific SCR it could merge with the specific SCR leading to a larger, combined response being erroneously measured (Fig 2C and 2D), or the NS-SCR might be the only response recorded if the SCL returned to baseline, and the subsequent specific SCR was disregarded (as only the first response was extracted in the event-related analysis). In some of these cases manual measurements were possible if the initial peak was easily identifiable, but if it was not clear where the true specific SCR started and peaked these data points had to be excluded. For the Epoch-analysis extracted data, if an artefact/NS-SCR just affected the $SCL_{base}$ value a simpler adjustment was often possible instead of manual remeasurement, by measuring the amplitude from the trough value at the start of the response ($SCL_{min}$) to the maximal response, $SCL_{max}$, provided that the $SCL_{min}$ was indeed the trough level before the peak, termed a 'max-min' switch. Manual remeasurement of SCRs was done in accordance with documented guidelines [37, 60, 61]. All manual inspection and remeasurement was performed by a researcher who was blinded to the subject's group and reviewed by an independent researcher. Non-responders were defined as participants in whom $\geq$50% raw SCRs following UCS presentations and pre-exposure tasks were <0.01μS, based on recommendations from previous literature [40, 62].

HR data was cleaned following manual inspection of ECG traces. Very few artefacts affected ECG data, mainly due to body movements, therefore only a small number of data required exclusion. Occasionally R-R intervals were incorrectly measured, if ECG complexes were small, in which case the detection threshold was changed and the analysis re-run to ensure all ECG complexes had detectable R-waves.

**Optimisation of data extraction method.** We compared both the automated Event-related analysis and Epoch analysis extraction methods to determine which was the most effective method for our data. In the Event-related analysis, the specific SCR is identified from the point at which the SCL first passes the 0.01μS threshold on the phasic SCL recording after the digital event marker of interest to the subsequent maximum peak SCL ($SCL_{max}$), shown in Fig 2. Using the data from four healthy volunteers, we reviewed all SCRs (288 trials) and scored manually for correct detection of SCRs. The Epoch analysis identified significantly more true SCRs than the Event-related analysis (50.3% vs. 34.4%, $\chi^2$ = 15.1; $P$ < .001). We applied both methods to the DMD and Control data from the second stage of the study. Artefacts and NS-SCRs were frequently present in the data and affected both extraction methods, and as such following manual inspection adjustments were required in 38.2% (423/1105) of Event-related analysis trials and 39.1% (417/1067) of Epoch analysis trials (Fig 3B). The data points for the two extraction methods were similar for both study cohorts when represented graphically (Fig 3A), although confidence intervals were wider in the event-related analysis in the Acquisition phase, suggesting lower accuracy for this method. The degree of data cleaning was greater for the Event-related extraction method, as time-consuming manual remeasurement was required in more than double the number of trials for the Event-related vs. Epoch method (38.2% vs. 16.7%; $\chi^2$ = 126.5; $P$ < .001). The more rapid 'max-min' switch was used in the remaining 22.3% of Epoch trial adjustments. Data exclusion rates were similar for both methods, with exclusion rates of approximately 9% for both methods in the familiarisation and acquisition phases, and in the extinction phase 18.0% for the epoch analysis and 20.0% for the event-related analysis, reflecting that in the majority of cases the artefacts leading to data exclusion would have affected the data no matter which extraction method was used. The epoch analysis raw data points were slightly closer to the cleaned data than the event-related analysis data points, although not significantly, which also suggests more accuracy with the epoch method (Fig 4). The cleaned SCR data points were very similar for both extraction methods, suggesting that there was good inter-method reliability for the independent manual inspection of the data.

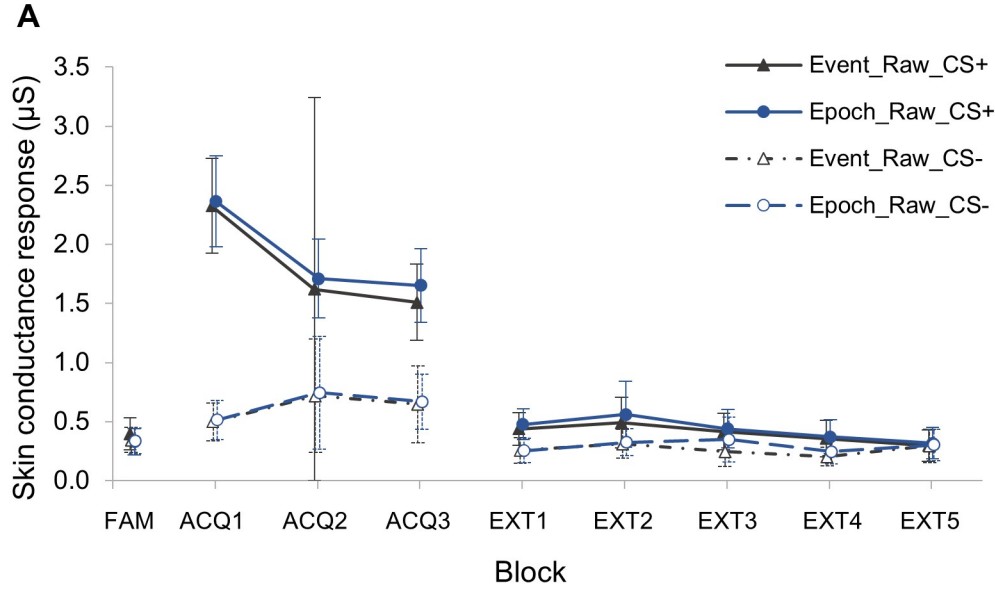

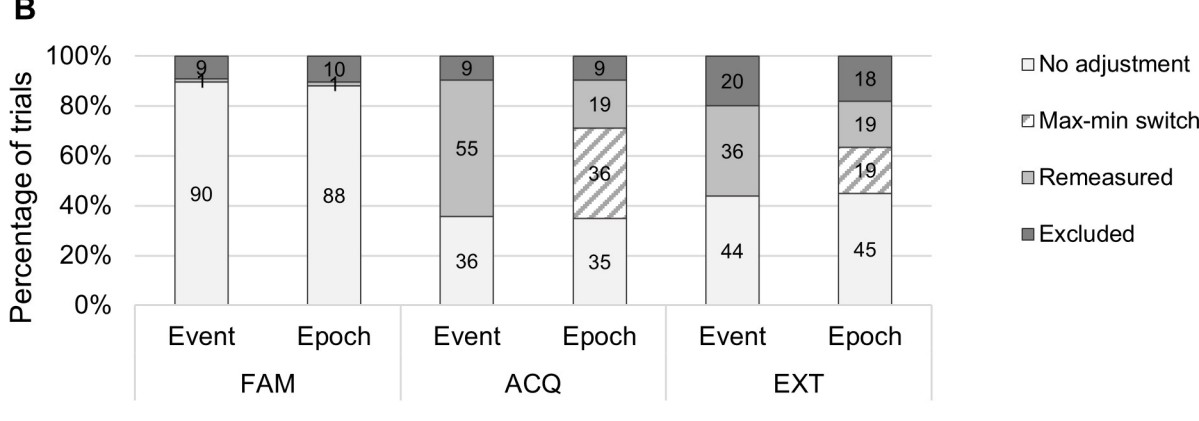

**Fig 3. Comparison of skin conductance response amplitude data derived by Event-related and Epoch analysis methods.** (**A**) Mean skin conductance responses (SCR) in μS from second stage of task development study (DMD and Control participants) in all phases of the fear conditioning paradigm (n = 17). SCRs are shown both for analysis using the epoch analysis extraction method ('Epoch'; circular markers) and the event-related analysis extraction method ('Event'; triangular markers), for both CS+ and CS- trials. Each data point represents the mean of four CS+ or CS- trials in each block, and error bars indicate 95% confidence intervals. 'Raw' data includes all data as extracted using these methods. 'Clean' data has excluded data points affected by artefacts, or data points that have been remeasured after manual inspection. FAM = Familiarisation phase; ACQ = Acquisition phase (1–3 blocks); EXT = Extinction (1–5 blocks). (**B**) Bar chart indicating percentages of trials included with and without adjustment, and those excluded following manual inspection. 'Max-min switch' refers to the use of the minimum skin conductance level ($SCL_{min}$) value within the trial window where it was more appropriate to use this than the SCL at the trial onset ($SCL_{base}$). 'Remeasured' refers to manual remeasurement, where artefacts prevented automated SCR measurement and a max-min switch was not appropriate or available.

Overall, we concluded that the Epoch analysis extraction method was superior to the Event-related analysis, given the more rapid and replicable data cleaning, less reliance on potentially subjective manual remeasurement and risks of unblinding and an indication of increased accuracy. It also allowed the SCR area and heart rate data to be extracted in the same process, thus improving efficiency.

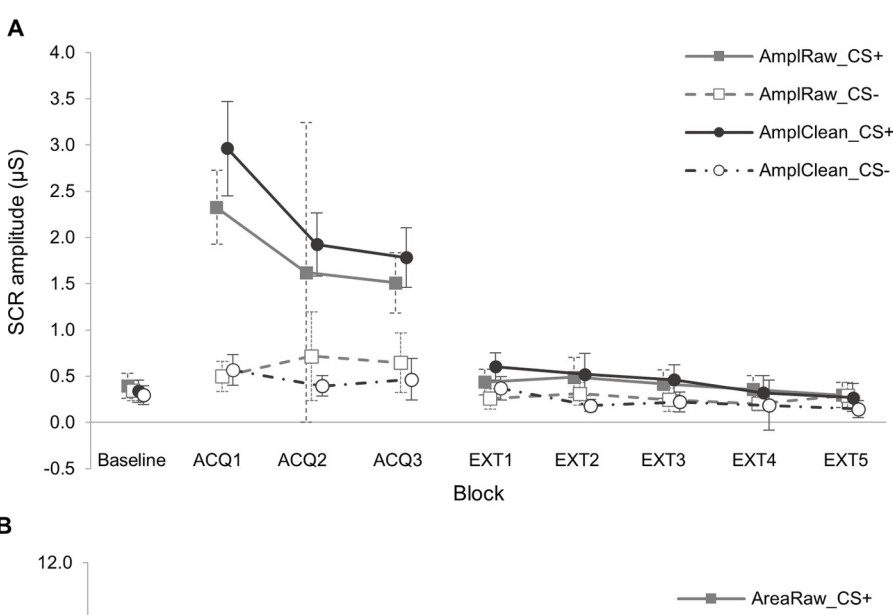

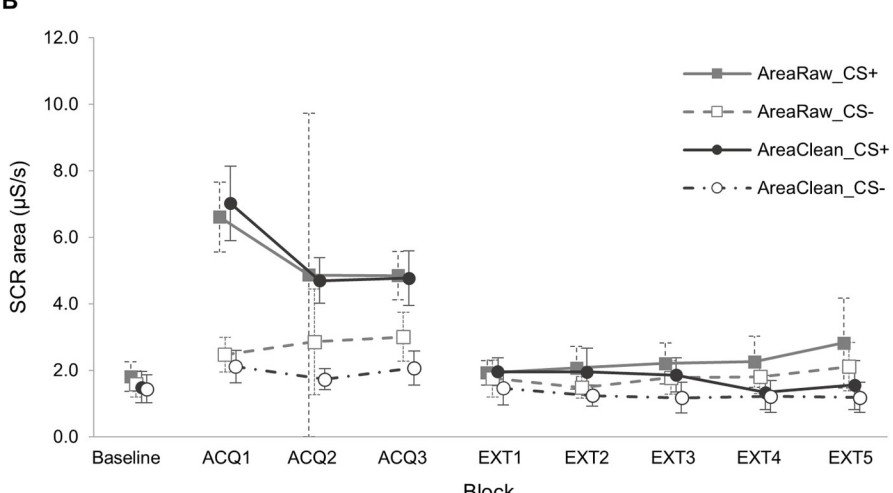

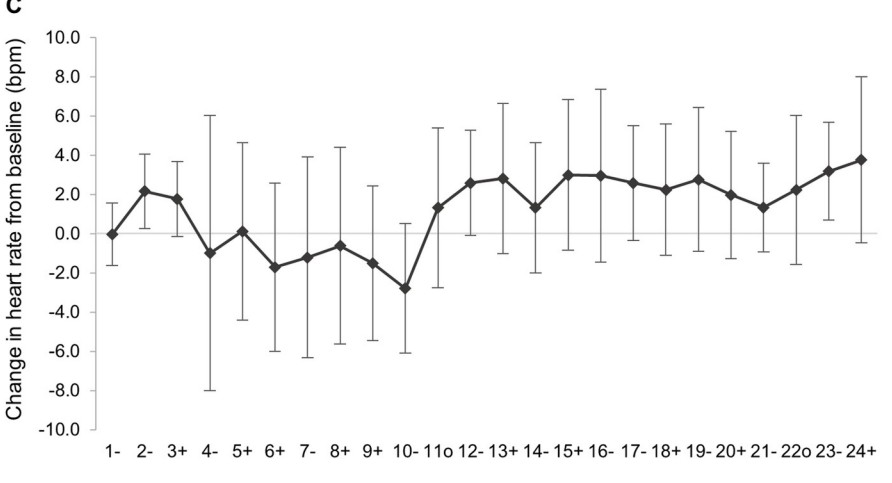

**Fig 4. Comparison of physiological startle response metrics.** (**A**) & (**B**) show comparison of raw and cleaned SCR data for all phases. (**A**) Mean skin conductance response amplitude (SCR$_{amp}$) for raw (AmplRaw) and cleaned (AmplClean) data by block. Unconditioned SCR startle response taken as the SCRamp on the first CS+ trial (termed 'SCR$_{uc}$'). (**B**) Mean skin conductance response area (SCR$_{area}$) for raw (AmplRaw) and cleaned (AmplClean) data by block. Baseline = baseline recordings taken from mean of Familiarisation phase SCR; ACQ = Acquisition phase (1–3

blocks); EXT = Extinction (1–5 blocks). (**C**) Change in heart rate from baseline (ΔHR) in Acquisition phase. Baseline heart rate (HR) taken from mean HR in Familiarisation phase. Data points are mean ΔHR for each numbered trial, where '+' denotes a CS+ trial, '-' denotes a CS- trial and and '°' indicates a non-reinforced CS+ trial, i.e. no unconditioned stimulus (UCS) presented. The unconditioned HR startle response metric was taken from the change in HR after first CS+ trial, i.e. from trial '3+' to trial '4-' (termed $\Delta HR_{uc}$). All data were obtained by Epoch analysis extraction. Error bars show 95% confidence intervals.

## Statistical considerations

**Sample size.** For the healthy volunteers study the sample size of six participants was selected for pragmatic reasons including time and resources constraints. There is no prior data from a DMD population to base a formal power calculation on. As a guide, a power calculation estimation was performed using healthy paediatric control SCR data from a discrimination fear conditioning study conducted by Gao and colleagues, with mean SCR 0.1μS and standard deviation 0.075μS for 8 year old subjects (n = 172) [31]. Taking 0.05μS as the precision required to detect a clinically significant difference, as in other paediatric studies [31, 63], and alpha = .05, the estimated sample size required for a power of 80% would be 56, with 28 in each group. Previous studies measuring fear responses in anxious/non-anxious children have used total sample sizes of: 35 (group size n = 17/18) [24]; 54 (group size n = 16/38) [33]; 60 (group size n = 8/11/19/16) [30]. Pragmatic considerations potentially limiting the DMD sample size include the fact that is a rare disease with a restricted age range in the inclusion criteria. Furthermore for the purposes of this task development study we were not investigating group differences, therefore we used a sample size of 20, including both DMD and Control subjects, in order to test proof-of-principle which could then be expanded to a larger study to test group differences if we were successful at this stage.

**Quantitative variables.** SCR amplitude is a well-established outcome measure for fear-conditioning studies, as the SCR response is typically a discrete peak in the EDA recording from which obtaining the amplitude is simple to measure both in automated analysis and manual measurements. Area under the curve has also been proposed a useful measure [64], as it captures the additional information of both the maximal amplitude and the duration of the response rather than just the maximum. A review of the initial data from the healthy volunteer study indicated that both SCR amplitude ($SCR_{amp}$) and area ($SCR_{area}$) were potentially useful SCR metrics. The heart rate (HR) responses to stimuli were continuous changes in HR rather than the discrete peaks seen for SCRs, therefore it was not possible to discriminate between CS + and CS- in the same was as for SCR, therefore a differential HR metric was not derived. Instead, the change in the absolute heart rate from the baseline heart rate (ΔHR) was calculated. The measured variables were: 1) $SCR_{amp}$ and $SCR_{area}$ in CS+ trials ($SCR_{CS+}$) and CS- trials ($SCR_{CS-}$); 2) Differential SCR, $SCR_{Diff}$, which is the difference between SCR from contiguous CS+ and CS- trials; 3) Absolute heart rate (HR); 4) Change in HR from baseline (ΔHR), where the baseline was mean HR in the baseline phase for each section of the task (familiarisation phase HR for acquisition trials; pre-extinction baseline phase for extinction trials). As our primary aim was to investigate startle responses, the main outcomes of interest were the initial unconditioned responses to the threat stimulus, which were the initial unconditioned SCR ($SCR_{uc}$) and the change in HR after the initial threat stimulus ($\Delta HR_{uc}$).

Normality checks using Q-Q plots showed approximately normal distributions for $SCR_{uc}$ and $\Delta HR_{uc}$, however on Shapiro-Wilks testing, $\Delta HR_{uc}$ failed the normality testing (test statistic for SCR: 0.92, df 18, $P$ = .13; test statistic for ΔHR: 0.86, 21 18, $P$ = .006), although the small group sizes may have made this assessment inaccurate.

**Data analysis.** Prior to analysis all subjects were randomly allocated a unique blinding identifier to allow blinded analysis to be conducted. All SCR and HR metrics were plotted and

visually inspected as an initial evaluation. For the healthy volunteer analysis, as the task protocol was modified after the first two participants, only the remaining four participants data was used for further comparison.

Paired sample *t*-tests were used to evaluate: task efficacy and degree of conditioning (response retention in extinction), based on the difference between $SCR_{CS+}$ and $SCR_{CS-}$; for any differences in $SCR_{uc}$ and $\Delta HR_{uc}$ between Visits 1 and 2. Coefficients of variation (CV) were used to compare outcome measures, where CV = (sd/mean)*100. Chi-squared tests were used to compare data extraction methods and pubertal stage, and baseline characteristics of the DMD and Control groups were compared with Mann-Whitney U tests. Spearman correlations and Bland-Altman plots and were used to evaluate test-retest reliability of $SCR_{uc}$ and $\Delta HR_{uc}$ metrics. Validity of Bland-Altman plots was confirmed with paired *t*-tests for each data set.

## Results

### Participant demographics

In the first task development stage, six participants took part in a healthy volunteers' study. The first two participants undertook the first iteration of the paradigm, following which a review of the data and feedback from these participants was used to make some modifications to the task. Four more participants, all male and aged between 20–25 years old, took part in the final version of the task. From an initial data review of the healthy0 volunteer data, we determined that the task achieved its purpose: it was able to elicit from participants appropriate and recordable physiological responses, particularly event-related skin conductance responses.

In the second stage of the task development, we recruited and tested 20 young DMD and Control participants aged 7–12 years old (DMD n = 11; Control n = 9). Out of these 20, SCR data could not be used in three participants (DMD n = 2, Control n = 1): one with technical difficulties in the EDA recording, one defined as a 'non-responder' and one with a protocol deviation (the participant stopped and re-started several times) so the data could not be used for either SCR or HR. The final analysis included SCR data from 17 participants (DMD n = 9, Control n = 8) and HR data from 19 participants (DMD n = 11, Control n = 8).

There was no difference in median age between the groups, and the distribution of pubertal stage scores also did not differ between groups (Table 1). The mean heart rate was significantly higher in the DMD group compared to Controls (*P* = .02), which is a recognised phenomenon in the literature [46], but there was no difference in baseline SCR between groups.

### Confirmation of task efficacy

Data from both the first healthy volunteer study and the study in young DMD and Control participants indicated the maximal SCR amplitude ($SCR_{amp}$) occurred on the first reinforced CS+ trial, i.e. the initial response to the threat stimulus, and also in the first acquisition phase block (Fig 4A), with significant discrimination between $SCR_{amp}$ in CS+ and CS- trials on first presentation of the threat stimulus, and throughout all Acquisition blocks. This was most marked in block one, ACQ1 (mean difference 2.4 µS (95%CI 1.6, 3.1); *P* < .001), with the largest difference occurring on the first reinforced CS+ trial (mean difference 4.1 µS (95% CI 2.6, 5.6); *P* < .001) (Table 2).

To confirm whether conditioning had occurred, we checked for the conditioned response retention when the threat stimulus was no longer presented in CS+ trials in the Extinction phase. There was significant discrimination between $SCR_{CS+}$ and $SCR_{CS-}$ in the first three Extinction blocks (*P* = .01; *P* = .02; *P* = .02; Table 2), indicating that the $SCR_{CS+}$ had been

**Table 1. Baseline demographics of DMD and Control participants.**

| | Whole cohort | | | | | DMD group[b] | | | | | Control group | | | | |
|---|---|---|---|---|---|---|---|---|---|---|---|---|---|---|---|
| **No. participants, n** | 20 | | | | | 11 | | | | | 9 | | | | |
| **Age, median** *(range)* | 10.2 (7.1–12.9) | | | | | 10.1 (7.1–11.4) | | | | | 10.4 (8.1–12.9) | | | | |
| **Pubertal stage score** | 1.0 | 1.5 | 2.0 | 2.5 | 3.0 | 1.0 | 1.5 | 2.0 | 2.5 | 3.0 | 1.0 | 1.5 | 2.0 | 2.5 | 3.0 |
| **Frequency**[a] *(%)* | 8/21 (38) | 6/21 (27) | 3/21 (14) | 2/21 (10) | 2/21 (10) | 5/12 (42) | 4/12 (33) | 2/12 (17) | 1/12 (8) | 0/12 (0) | 3/9 (33) | 2/9 (22) | 1/9 (11) | 1/9 (11) | 2/9 (22) |
| **Baseline SCR, median** *(range)* | 0.23 (0.0–0.76) | | | | | 0.23 (0.01–0.76) | | | | | 0.23 (0.0–0.49) | | | | |
| **Baseline HR, median** *(range)* | 91.3 (70.2–126.5) | | | | | 95.4 (70.1–126.5); *P* = **.02**[c] | | | | | 80.1 (70.1–102.1) | | | | |

Data presented for the whole cohort (n = 20) as well as DMD (n = 11) and Control groups (n = 9). Median and range presented for variables: age, skin conductance response (SCR); heart rate (HR).

[a]Pubertal stage score data shown as frequencies (%) in each category (ranging from 1–5).

[b]Group differences were assessed with Mann-Whitney U-tests (age, baseline SCR and HR) and Chi-squared test (pubertal stage), with alpha level *P* = .05.

[c]significant difference in baseline HR between DMD and Control groups; *P* = .02.

conditioned as it remained higher than $SCR_{CS-}$. This significant discrimination was lost for the remaining two Extinction blocks, indicating that the conditioned response had extinguished.

We have thus confirmed that this new fear-conditioning task achieved the aim of producing physiological responses to the unconditioned stimulus, and successfully conditioning these responses, which could then be extinguished. It also supported the extended Extinction phase that we employed based on preliminary data from the healthy volunteer study.

## Evaluation of physiological variables

**Comparing SCR amplitude and area metrics.** As well as measuring SCR amplitude ($SCR_{amp}$) as described above, we also investigated using the area under the phasic skin conductance response curve ($SCR_{area}$) as an additional exploratory measure, as theoretically this could provide temporal as well as magnitude information [37]. Raw and cleaned data for $SCR_{amp}$ and $SCR_{area}$, shown in Fig 4, indicate that the $SCR_{area}$ metric appeared more accurate than $SCR_{amp}$ for CS+ trial data in Acquisition, but the discrepancy between raw and cleaned data

**Table 2. Comparison of skin conductance responses in paired CS+ and CS- trials.**

| $SCR_{CS+}$ vs. $SCR_{CS-}$ | N | Mean difference (µS) | 95% CI | t | Sig., P |
|---|---|---|---|---|---|
| **1st CS+/CS- ACQ trial pair** | 16 | 4.1 | 2.6, 5.6 | 5.7 | **< .001** |
| **ACQ block 1** | 18 | 2.4 | 1.6, 3.1 | 6.9 | **< .001** |
| **ACQ block 2** | 17 | 1.7 | 0.9, 2.5 | 4.5 | **< .001** |
| **ACQ block 3** | 16 | 1.4 | 0.9, 2.0 | 6.0 | **< .001** |
| **EXT block 1** | 17 | 0.31 | .08, 0.54 | 2.9 | **.01** |
| **EXT block 2** | 15 | 0.32 | 0.05, 0.59 | 2.5 | **.02** |
| **EXT block 3** | 13 | 0.28 | 0.05, 0.51 | 2.6 | **.02** |
| **EXT block 4** | 12 | 0.17 | -0.10, 0.41 | 1.3 | .20 |
| **EXT block 5** | 11 | 0.16 | -0.01, 0.32 | 2.2 | .06 |

Skin conductance responses (SCR) in CS+ and CS- trials were compared for the first trial pair in Acquisition phase, and for the mean of all CS+ and CS- trials in each block (four trial pairs per block) during Acquisition (ACQ) and Extinction (EXT) phases. Statistical comparisons were performed with paired *t*-tests. CS = conditioned stimulus; CI = confidence interval; Sig. = significance, with alpha level = .05. *P*-values < .05 are shown in bold.

was greater in the Extinction phase for $SCR_{area}$ than $SCR_{amp}$. However, data cleaning was more time-consuming and potentially subjective and prone to bias for area measurements, due to the higher chance of artefacts and NS-SCR affecting the data within the whole sampling window. Due to these limitations, we concluded that future analysis should use SCR amplitude as the primary SCR metric.

**Heart rate metrics: Absolute heart rate and change in heart rate.** As has previously been established, the mean heart rate was significantly higher in the DMD group compared to the Control group (Fig 5B), therefore evaluation of absolute heart rate between groups was not a useful comparison. It was also not possible to derive a 'differential' heart rate, comparing CS + and CS- trials as HR varied in a continuous way rather than in discrete peaks as seen with SCR, therefore it was more informative to represent the change in HR (ΔHR) compared to baseline, showing all CS+ and CS- trial data in the order presented in the task (Fig 5C).

**Unconditioned startle response outcome measures.** As our hypothesis concerned measurement of startle responses, we evaluated the unconditioned physiological response metrics to determine the optimal outcome measures for the startle response; firstly how to derive them, and secondly how precise and reliable they were.

As the greatest difference in $SCR_{amp}$ between CS+ and CS- trials occurred at the start of the Acquisition phase, we used the SCR amplitude in the first CS+ trial as the unconditioned SCR metric (termed '$SCR_{uc}$'). For the optimal HR metric, from inspection of the ΔHR data we determined that there was a reduction in HR after each reinforced CS+ trial in the early stages of the Acquisition phase (Fig 4C), which was largest after the first reinforced CS+ trial. This is a recognised phenomenon known as fear-induced bradycardia [42]. We measured the difference in HR after the first reinforced CS+ trial and took this as the unconditioned HR metric (termed '$ΔHR_{uc}$').

In Fig 4C it was also evident from the wide confidence intervals for the ΔHR data point after the first CS+ trial (trial 4-) that there was a large variation in $ΔHR_{uc}$, which might reduce the utility of this as an outcome measure. To determine this more precisely, we calculated coefficients of variation (CV) for each measure. A CV of 100% occurs when the standard deviation of a measure is equal to its mean, and as CV reduces below 100% towards 0% this indicates increasing precision [65]. The CV for $SCR_{uc}$ was 67.8%, the CV for SCRs in CS+ trials in the first Acquisition block was 56.7% and the CV for $ΔHR_{uc}$ was 103.1%, indicating that the SCR metrics were more precise than the $ΔHR_{uc}$.

To determine test re-test reliability of these outcomes, eleven DMD participants were followed-up at a second visit after an interval of three months. We obtained eleven sets of paired HR data and seven sets of paired SCR data, although one set of SCR data had to be excluded as the participant was a 'non-responder' on both occasions (Fig 5). There was a strong positive correlation in $SCR_{uc}$ between visit 1 and 2 (Spearman's *rho* = .86, *P* = .01, n = 6), thus meeting our pre-determined criteria for a correlation coefficient of ≥0.8. There was no difference in mean $SCR_{uc}$ between visits 1 and 2 (mean difference 0.19 μS; 95% confidence interval -1.2,1.6; *t* = .35, *P* = .73), or for the remainder of the CS+ and CS- trials in Visit 1 compared to Visit 2 on paired *t*-tests. Individual paired $SCR_{uc}$ data appeared relatively stable values for most participants between visits (Fig 5B).

There was no difference in $ΔHR_{uc}$ between visits (mean difference -2.9 bpm; 95% confidence interval -9.6, 3.9; *t* = -.95; *P* = .37) (Fig 5C), however, there was more individual variability in this metric (demonstrated on paired data in Fig 5D), and only a moderate, non-significant positive correlation within participants between visits (*rho* = .42, *P* = .20).

To further explore reliability, Bland-Altman Limits of Agreement (LOA) plots of difference vs. mean were constructed for both data sets (Fig 5E and 5F). For both $SCR_{uc}$ and $ΔHR_{uc}$ all data points were within the LOA boundaries, indicating agreement in both outcomes on test-

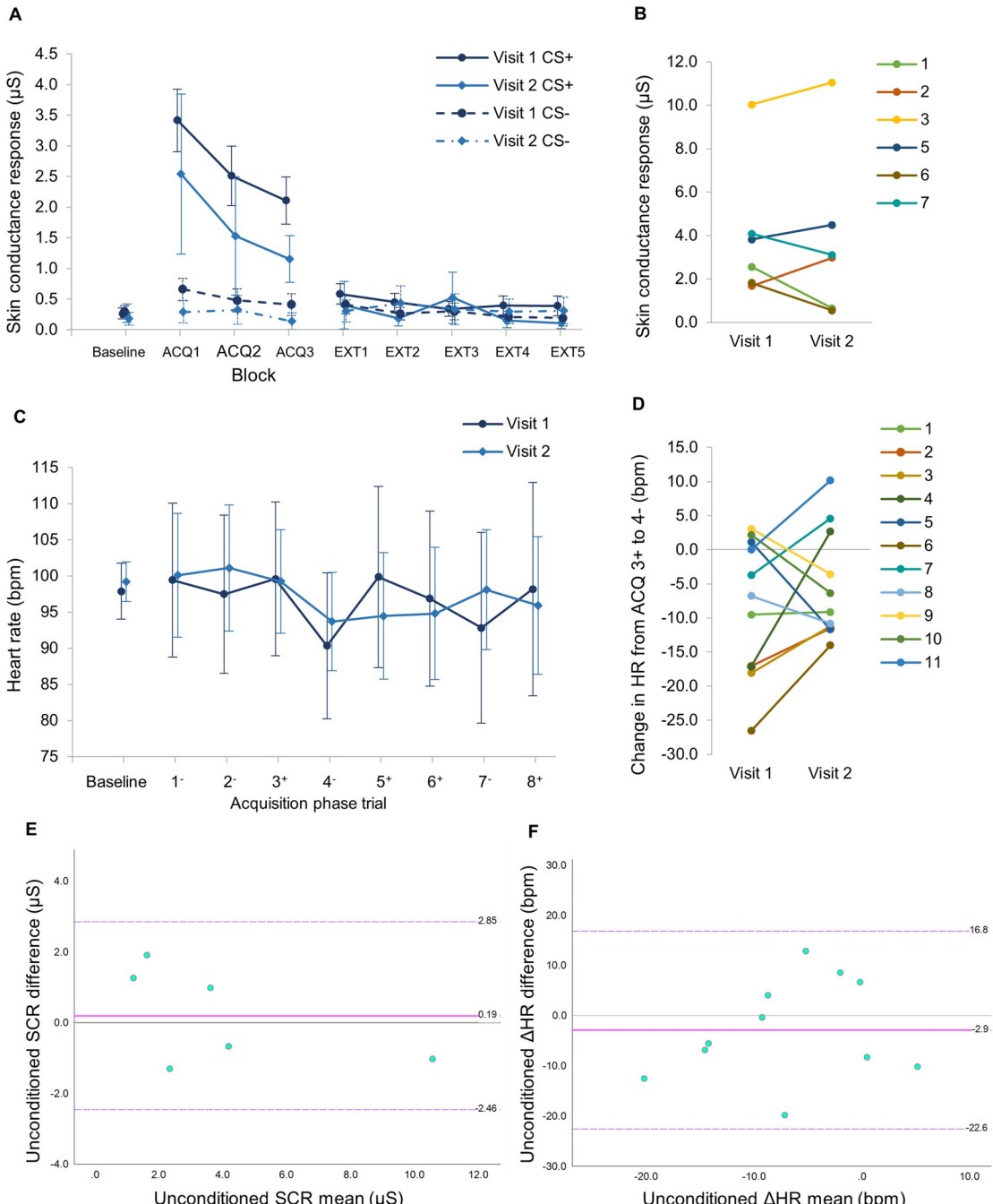

**Fig 5. Comparison of SCR and HR responses in DMD participants at initial and follow-up visits.** Data for n = 11 DMD participants who performed the fear-conditioning task at the initial visit (Visit 1)and at a second follow-up visit after 3 months (Visit 2). Paired HR data available for n = 11 and paired SCR data available for n = 6. (**A**) Mean SCR by block for CS+ and CS- trials over all task phases. Error bars show 95% confidence intervals. (**B**) Paired data for individual participants showing the SCR for the first unconditioned response (SCR$_{uc}$) in Visit 1 and Visit 2. (**C**) Mean absolute HR by trial in the first Acquisition block. Error bars show 95% confidence intervals. (**D**) Paired data for individual participants showing the Change in HR (ΔHR) from Acquisition trial 3+ to trial 4-, i.e. the unconditioned ΔHR response (ΔHR$_{uc}$). (**E**) & (**F**). Bland-Altman Limits of Agreement plots for: (**E**) Unconditioned skin conductance response (SCR; n = 6) and (**F**) Unconditioned change in heart rate (ΔHR; n = 11). Plots show the mean difference (solid pink line) and upper and lower limits of agreement, LOA, in dashed purple lines. Data labels indicate the values for upper LOA, mean difference and lower LOA (from top to bottom). All data points for both SCR$_{uc}$ and ΔHR$_{uc}$ lie within these LOA boundaries, indicating there is agreement between the test-retest outcomes.

retest assessment. However, given the small sample sizes, further study should be done to confirm this.

Overall, these results show that $SCR_{uc}$ is a reliable outcome, showing a strong within-subjects correlation at retest, agreement on LOA plot and less variance than $\Delta HR_{uc}$. $\Delta HR_{uc}$ also did not show significant test-retest correlation. Therefore, we conclude that $SCR_{uc}$ is the more useful outcome measure for this task.

## Discussion

This study presents the development and initial evaluation of a novel fear conditioning task in both healthy volunteers and a clinical population of young males with Duchenne muscular dystrophy and age-matched Control participants. We have shown that the task is effective, have optimised the data processing and analysis methods and have determined physiological startle response metrics which could be studied further as a potential outcome measures in testing the emotion system in children and youth with DMD.

These results confirm that the experimental protocol is effective in eliciting physiological unconditioned startle responses, with an increase in skin conductance response, SCR, and decrease in heart rate, HR, following presentation of an aversive stimulus. The SCRs were successfully conditioned to a neutral stimulus, and the conditioned SCRs subsequently extinguished with repeated presentation of the neutral stimulus alone. Thus, this experimental paradigm meets the objectives of the task design.

After evaluating of physiological outcome measures, we found that the optimal outcome measure in this paradigm is SCR amplitude. SCR area was effective, but its measurement was prone to more errors and required more data processing and cleaning. By comparing two data extraction methods, we determined the optimal processing pathway for SCR amplitude measurement. We also showed that the unconditioned SCR ($SCR_{uc}$) metric was reliable on test-retest analysis and was much more precise than the unconditioned change in heart rate $\Delta HR_{uc}$ metric.

From the heart rate variables, absolute HR and change in HR ($\Delta HR$), $\Delta HR$ was the preferred metric due to the group differences in baseline HR between DMD and Control groups. The specific unconditioned $\Delta HR$ metric, $\Delta HR_{uc}$, was used to determine the presence of post-threat bradycardia, which is a recognised parasympathetic nervous system response in fearful situations, analogous to the 'freeze' response seen in animals [42]. However, this metric showed greater variance than $SCR_{uc}$ and did not show significant correlation on test-retest analysis. A further limitation of using heart rate metrics in this context is that DMD is associated with a resting sinus tachycardia from an early age, and decreased heart rate variability as the condition progresses, thought to be related to dysautonomia [46, 66]. This may limit the useful comparison of HR metrics between DMD and Control groups. However, HR may provide a useful within-subject outcome measure, as the task elicited a measurable unconditioned response, and recent studies in humans have suggested it can provide a reliable outcome measure [42]. Further work using a larger data set should enable a more detailed analysis to determine the optimal HR metric.

The main limitations in this study were related to artefacts and data drop-out. The occurrence of physiological artefacts due to external noises may have been reduced if testing had occurred in a soundproof room, which was not available to us for this study although we tried to mitigate this with the use of headphones. Some participants who terminated both phases of the task early (20% in acquisition, 25% in extinction), mostly DMD participants, which led to reduced data especially in the Extinction phase. Feedback from participants indicated that the task was too long, and a minority of participants (mostly in the DMD group) found the noise

stimulus too aversive. Therefore, future versions of the task should incorporate modifications to reduce the length of the task and incorporate modified stimuli that will balance aversiveness with the ability to elicit physiological startle responses and produce successful conditioning.

In conclusion, this task development study has achieved its aims by confirming that this novel fear-conditioning task is effective in eliciting unconditioned startle responses and successful conditioning and establishing the optimal outcome measures and data processing procedures. These can be applied to future larger studies investigating group differences in startle responses between DMD and Control participants.

## Supporting information

**S1 File. Summary of previously published study designs for fear-conditioning tasks in children/youths.** This table presents study protocol details obtained from nine previously published studies using fear-conditioning tasks with paediatric participants, which informed some aspects of the development of the novel task we created for this study. References: [1]Gao Y, Raine A, Venables PH, Dawson ME, Mednick SA. The development of skin conductance fear conditioning in children from ages 3 to 8 years. Dev Sci. 2010;13(1):201–12.; [2]Pattwell SS, Duhoux S, Hartley CA, Johnson DC, Jing D, Elliott MD, et al. Altered fear learning across development in both mouse and human. Proc Natl Acad Sci USA. 2012;109(40):16318–23.; [3]Neumann DL, Waters AM, Westbury HR, Henry J. The use of an unpleasant sound unconditional stimulus in an aversive conditioning procedure with 8- to 11-year-old children. Biol Psychol. 2008;79(3):337–42.; [4]Shechner T, Britton JC, Ronkin EG, Jarcho JM, Mash JA, Michalska KJ, et al. Fear conditioning and extinction in anxious and nonanxious youth and adults: examining a novel developmentally appropriate fear-conditioning task. Depress Anxiety. 2015;32(4):277–88.; [5]Field AP. I don't like it because it eats sprouts: conditioning preferences in children. Behav Res Ther. 2006;44(3):439–55.; [6]Lau JY, Lissek S, Nelson EE, Lee Y, Roberson-Nay R, Poeth K, et al. Fear conditioning in adolescents with anxiety disorders: results from a novel experimental paradigm. J Am Acad Child Adolesc Psychiatry. 2008;47 (1):94–102.; [7]Glenn CR. Comparing electric shock and a fearful screaming face as unconditioned stimuli for fear learning. Int J Psychophysiol. 2012;86(3):214–9.; [8]Jovanovic T, Nylocks KM, Gamwell KL, Smith A, Davis TA, Norrholm SD, et al. Development of fear acquisition and extinction in children: effects of age and anxiety. Neurobiol Learn Mem. 2014;113:135–42.; [9]Schiele MA, Reinhard J, Reif A, Domschke K, Romanos M, Deckert J, et al. Developmental aspects of fear: Comparing the acquisition and generalization of conditioned fear in children and adults. Dev Psychobiol. 2016;58(4):471–81.
(PDF)

**S2 File.**
(PDF)

## Acknowledgments

We would like to acknowledge the support of the National Institute of Health Research (NIHR) Biomedical Research Centre at Great Ormond Street Hospital for Children NHS Foundation Trust. The views expressed are participants of the author(s) and not necessarily participants of the NHS, the NIHR or the Department of Health. We thank David Watson (University of Sussex), Charles Marshall (UCL) & Rachael Elward (UCL) for their technical advice in scripting the fear-conditioning task. We gratefully acknowledge the GOSH Somers Clinical Research Facility, the participation of Muscular Dystrophy UK, Duchenne UK and

Action Duchenne in promoting study recruitment and all the study participants and their parents/carers.

## Author Contributions

**Conceptualization:** Kate Maresh, Neil Harrison, William Mandy, David Skuse, Francesco Muntoni.

**Data curation:** Kate Maresh, Andriani Papageorgiou.

**Formal analysis:** Kate Maresh, Andriani Papageorgiou, Deborah Ridout.

**Funding acquisition:** Kate Maresh, Francesco Muntoni.

**Investigation:** Kate Maresh, Andriani Papageorgiou.

**Methodology:** Kate Maresh, Andriani Papageorgiou, Deborah Ridout, Neil Harrison, William Mandy, David Skuse, Francesco Muntoni.

**Project administration:** Kate Maresh.

**Resources:** Kate Maresh, Francesco Muntoni.

**Supervision:** Deborah Ridout, William Mandy, David Skuse, Francesco Muntoni.

**Writing – original draft:** Kate Maresh, Andriani Papageorgiou.

**Writing – review & editing:** Kate Maresh, Andriani Papageorgiou, Deborah Ridout, Neil Harrison, William Mandy, David Skuse, Francesco Muntoni.

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
