## [Decision Letter · Decision Letter 0]

3 Feb 2022

Development of a novel startle response task in Duchenne muscular dystrophy

PONE-D-21-31296

Dear Dr. Muntoni,

We’re pleased to inform you that your manuscript has been judged scientifically suitable for publication and will be formally accepted for publication once it meets all outstanding technical requirements.

Kind regards,

Sayamwong E. Hammack, Ph.D

Academic Editor

PLOS ONE

Additional Editor Comments (optional):

Reviewers' comments:

Reviewer's Responses to Questions

**Comments to the Author**

1. Is the manuscript technically sound, and do the data support the conclusions?

Reviewer #1: Yes

2. Has the statistical analysis been performed appropriately and rigorously? 

Reviewer #1: I Don't Know

3. Have the authors made all data underlying the findings in their manuscript fully available?

Reviewer #1: Yes

4. Is the manuscript presented in an intelligible fashion and written in standard English?

Reviewer #1: Yes

5. Review Comments to the Author

Reviewer #1: This study evaluates the startle response task to evaluate DMD patients. The objective of the study is to validate the task as a novel tool for measuring emotion system / outcome of DMD patients. The study was well designed and carried out carefully. I believe the study can be published as is.

6. PLOS authors have the option to publish the peer review history of their article (what does this mean?). If published, this will include your full peer review and any attached files.

Reviewer #1: No

---

## [Editor Report · Acceptance letter]

4 Apr 2022

PONE-D-21-31296 

Development of a novel startle response task in Duchenne muscular dystrophy 

Dear Dr. Muntoni:

I'm pleased to inform you that your manuscript has been deemed suitable for publication in PLOS ONE. Congratulations! Your manuscript is now with our production department. 

Kind regards, 

on behalf of

Dr. Sayamwong E. Hammack 

Academic Editor

PLOS ONE